# Chromosomal evolution and phylogenetic considerations in cuckoos (Aves, Cuculiformes, Cuculidae)

**Michelly da Silva dos Santos**[1,2‡], **Rafael Kretschmer**[3‡], **Ivanete de Oliveira Furo**[1,2], **Ricardo José Gunski**[4], **Analía del Valle Garnero**[4], **Mirela Pelizaro Valeri**[5], **Patricia C. M. O'Brien**[6], **Malcolm A. Ferguson-Smith**[6], **Edivaldo Herculano Corrêa de Oliveira**[1,2,7]*

**1** Programa de Pós-graduação em Genética e Biologia Molecular, PPGBM, Universidade Federal do Pará, Belém, Pará, Brazil, **2** Laboratório de Cultura de Tecidos e Citogenética, SAMAM, Instituto Evandro Chagas, Ananindeua, Pará, Brazil, **3** Programa de Pós-graduação em Genética e Biologia Molecular, PPGBM, Universidade Federal do Rio Grande do Sul, Porto Alegre, Rio Grande do Sul, Rio Grande do Sul, Brazil, **4** Programa de Pós-graduação em Ciências Biológicas, PPGCB, Universidade Federal do Pampa, São Gabriel, Rio Grande do Sul, Rio Grande do Sul, Brazil, **5** Departamento de Genética, Ecologia e Evolução, Laboratório de Citogenômica Evolutiva, Instituto de Ciências Biológicas, Universidade Federal de Minas Gerais, Belo Horizonte, Minas Gerais, Brazil, **6** Department of Veterinary Medicine, Cambridge Resource Centre for Comparative Genomics, University of Cambridge, Cambridge, United Kingdom, **7** Instituto de Ciências Exatas e Naturais, Universidade Federal do Pará, Belém, Pará, Brazil

‡ These authors are joint first authors and contributed equally to this work.
* ehco@ufpa.br

**Data Availability Statement:** All relevant data are within the paper.

## Abstract

The Cuckoos have a long history of difficult classification. The species of this order have been the subject of several studies based on osteology, behavior, ecology, morphology and molecular data. Despite this, the relationship between Cuculiformes and species of other orders remains controversial. In this work, two species of Cuculidae, *Guira guira* (Gmelin, 1788) and *Piaya cayana* (Linnaeus, 1766), were analyzed by means of comparative chromosome painting in order to study the chromosome evolution of this group and to undertake the first chromosome mapping of these species. Our results demonstrate high chromosomal diversity, with 2n = 76 in *G. guira*, with fission and fusion events involving ancestral syntenies, while *P. cayana* presented only fissions, which were responsible for the high diploid number of 2n = 90. Interestingly, there were no chromosomal rearrangements in common between these species. Our results, based on Giemsa staining, were compared with previous data for other cuckoos and also with taxa proposed as sister-groups of Cuculiformes (Otidiformes, Musophagiformes and Opisthocomiformes). Cytogenetic comparisons demonstrated that cuckoo species can be divided into at least three major groups. In addition, we found no evidence to place Cuculiformes close to the groups proposed previously as sister-groups.

## Introduction

The order Cuculiformes is represented by only one family, Cuculidae (cuckoos), which is an ancient and diverse group comprising 149 species [1]. Despite the fact that the majority of

**Funding:** This work was supported by the Conselho Nacional de Desenvolvimento Científico e Tecnológico (CNPq) Grant number 307382/2019-02 to EHCO and the Coordenação de Aperfeiçoamento de Pessoal de Nível Superior (CAPES). The funders had no role in study design, data collection and analysis, decision to publish, or preparation of the manuscript.

**Competing interests:** The authors have declared that no competing interests exist.

species of cuckoos build their own nests, sometimes breeding in a cooperative way, this group of birds is known for the cases of brood parasites [2,3]. Recently, cuckoos have been the subject of interesting studies, gaining attention in such different areas as the maternal inheritance of eggs [4], their evolution [3,5], biogeography [6] and phylogenetic relationships [7].

The phylogenetic relationships of cuckoos have a long and enigmatic history [8]. For instance, the hoatzin (*Opisthocomus hoazin* Müller, 1776) [9] and turacos (Musophagidae) [10] were considered as sister groups to cuckoos. However, molecular studies have not produced any evidence to consider the hoatzin as a Cuculiformes, or as their sister group [11]. Additionally, these authors found no evidence to support the close relationship between turacos and cuckoos or between turacos and hoatzin. Hence, despite the great progress in the reconstruction of the phylogeny of Aves in recent years, the position of these three groups remains unresolved. In fact, the most recent avian phylogenies suggest different sister relationships to Cuckoos: Prum et al. [12] support the bustards (Otidiformes) and Jarvis et al. [13], the turacos (Musophagiformes). Hence, despite these efforts, further studies are still required to resolve the evolutionary history and relationships of the Cuculiformes with other bird groups.

Up to now, there have been few cytogenetic studies of representative species of the Cuculiformes, and those that are available date from before 1991 and are mainly limited to a description of the diploid number (2n) and the karyotype [14–19]. These studies demonstrate a large variety of karyotypes, with diploid numbers ranging from 2n = 64 in *Crotophaga major* (Linnaeus, 1758) [15] to 2n = 76 in *Piaya cayana* (Linnaeus, 1766) [14].

However, in the last two decades, a large amount of data on chromosome mapping in birds has been generated [20], including in the hoatzin [21], which make genome comparison possible between Cuculiformes species. Hence, in this study we aim to investigate chromosome diversification in cuckoos, considering the challenges faced by other efforts to reconstruct the evolutionary processes that have occurred in this group of birds. Thus, for the first time, we describe whole chromosome painting in two species of cuckoos, the Guira Cuckoo (*Guira guira* Gmelin, 1788) and the Squirrel Cuckoo (*P. cayana*). Our results are compared with the chromosome painting data from the hoatzin [21] and the classical cytogenetics data from Otidiformes and Musophagiformes species, in order to check if the phylogeny proposed by Jarvis et al. [13] or that suggested by Prum et al. [12] is supported by the cytogenetic data.

## Material and methods

### Biological samples

Two species of cuckoos (Cuculiformes, Cuculidae) were analyzed in this study: the Guira Cuckoo (*G. guira*) and the Squirrel Cuckoo (*P. cayana*). The individuals were captured with mist nets in their natural environment (Table 1) and the experiments were approved by the Ethics Committee on Animal Experimentation (CEUA) of the Universidade Federal do Pampa under no. 026/2012. For *P. cayana*, birds were held manually, and an area of the underside of the wing was cleaned with ethanol 70%. Afterwards, lidocaine gel was applied for local

**Table 1. Characterization of the samples used in this study.**

| Species | Sample | 2n | Locality |
|---------|--------|-----|----------|
| *G. guira* | 2 Male | 76 | Santa Maria, Rio Grande do Sul State, Brazil |
| *P. cayana* | 1 Female | 90 | Santa Maria, Rio Grande do Sul State, Brazil |
| *P. cayana* | 1 Female | 90 | Porto Vera Cruz, Rio Grande do Sul State, Brazil |

anesthesia. A small biopsy (0,3 cm$^2$) was collected using a scalpel blade, and a topical antibacterial was applied before releasing the animal. For *G. guira*, animals were euthanatized using ketamine/xylazine, and the bone marrow was obtained from the femur.

## Chromosome preparation

Chromosomes were obtained either by short-term bone marrow [22] or fibroblast culture [23]. Both protocols included colchicine (0,05%) treatment for 1 h, hypotonic treatment (75 mM KCl) for 15 minutes, and cell fixation in methanol:acetic acid (3:1, v/v). After three to four rounds of washing/fixation, chromosome suspensions were kept in a freezer.

## Conventional staining and karyotype analyses

Chromosome suspensions were dropped onto a clean glass slide and stained with Giemsa in order to assess the morphology of macrochromosomes and to determine the chromosome number. The chromosome number was determined analyzing at least 20 metaphases using a 100× objective (Leica DM1000).

## Fluorescence *in situ* hybridization (FISH)

FISH experiments were performed using *Gallus gallus* probes corresponding to GGA1–10 and *Leucopternis albicollis* (LAL) probes homologous to GGA1 (LAL 3, 6, 7, 15, and 18), 2 (LAL 2, 4, and 20), 3 (LAL 9, 13, 17, and 26), 4 (LAL 1 and 16), 5 (LAL 5) and 6 (LAL 3) [24], labeled by biotin and detected using streptavidin-CY3 (Invitrogen). Both sets of probes were obtained by flow cytometry at the Cambridge Resource Centre for Comparative Genomics (Cambridge, UK). Fluorescence results were analyzed and acquired using a Zeiss Imager 2 microscope, 63× objective and Axiovision 4.8 software (Zeiss, Germany).

## Results

### Karyotype description

The karyotype of *G. guira* showed 76 chromosomes. The karyotype consists of 12 pairs of macrochromosomes: 3, 4, 7, 8, 9, 11 and 12 are metacentric; 1 and 2 are submetacentric; 5 is acrocentric and 6 and 10 are telocentric. The Z sex chromosome is submetacentric and its size lies between pairs 3 and 4. The W sex chromosome was not analyzed in this study (Fig 1).

*P. cayana* had 2n = 90. Macrochromosomes (13 pairs) of *P. cayana* are different in morphology and size from those of *G. guira* (Fig 2). Pairs 1, 5, 6 and 10 are telocentric; pair 2 is submetacentric; 3, 4 and 11 metacentric and the pairs 7, 8 and 9 are acrocentric. The sex chromosome Z is submetacentric and equivalent in size to pair 2. The W is acrocentric and equivalent in size to pair 7.

### FISH experiments

Whole-chromosome probes GGA1–10 showed that only the chromosomes GGA1, 2, 3 and 9 were not involved in interchromosomal rearrangements in *G. guira* (Figs 3 and 5A). Chicken chromosomes GGA6, 7 and 8 probes hybridized to two chromosomes each, while GGA4 and 5 probes hybridized to three pairs each. Chromosome fusions were found in four GGU chromosomes: GGA7seg/GGA4seg (GGU4), GGA5seg/GGA10 (GGU5), GGA5seg/GGA6seg (GGU8), GGA8seg/GGA4p (GGU9). Moreover, the GGU7q, GGU12p do not have homology to chromosomes GGA1-10, suggesting that these regions must indicate fusion with microchromosomes. Additionally, it was clear that the segment corresponding to GGA1q (GGU1p) was not hybridized by any probe from LAL, suggesting that there must be at least one more

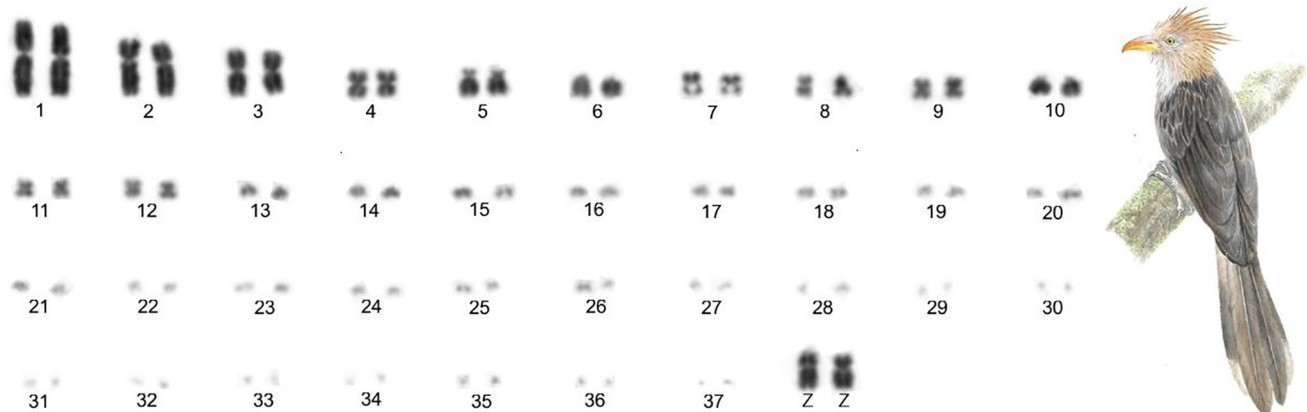

**Fig 1. Complete karyotype of a male *G. guira* (2n = 76), in conventional Giemsa staining.**

pair of LAL chromosomes corresponding to GGA1 that is missing from the current set of LAL probes, as pointed out by Kretschmer et al. [25]. Although the GGU1 and 3 are homologous to GGA1 and GGA3, respectively, the chromosome morphology of these chromosomes in GGU and the pattern of signals observed with LAL probes suggest the occurrence of centromere repositioning due to inversions or centromeric shift in both chromosomes. A homology map based on the results of the experiments using GGA and LAL probes in GGU is shown in Fig 5A.

The patterns of hybridization of the chicken paints GGA1–GGA9 to chromosomes of *P. cayana* indicate that GGA5, 6, 8, 9 and 10 are entirely homologous to PCA4, 7, 9, 11 and 12, respectively. Probes 1, 3 and 4 painted 2 distinct pairs of *P. cayana* each, while the syntenic group corresponding to GGA2 is split between three pairs (Figs 4 and 5B). The paint GGA7 indicates that this chromosome was involved in a fusion event (PCA7), probably with a micro-chromosome, since none of the GGA probes hybridized to a segment in the long arm of this chromosome. *L. albicollis* probes confirmed the results obtained with *G. gallus* probes.

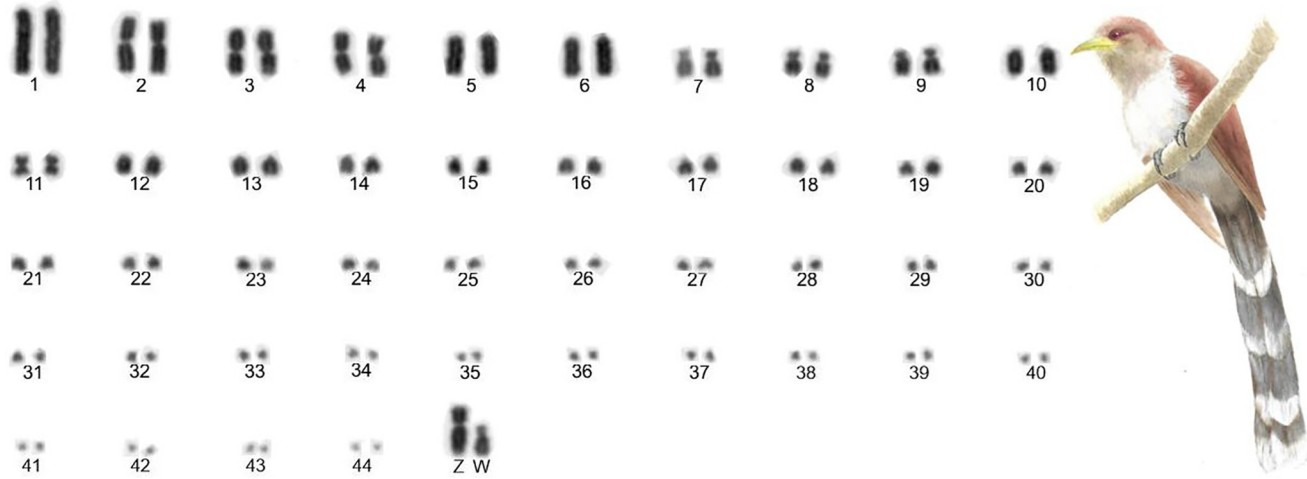

**Fig 2. Complete karyotype of a female *P. cayana* (2n = 90), in conventional Giemsa staining.**

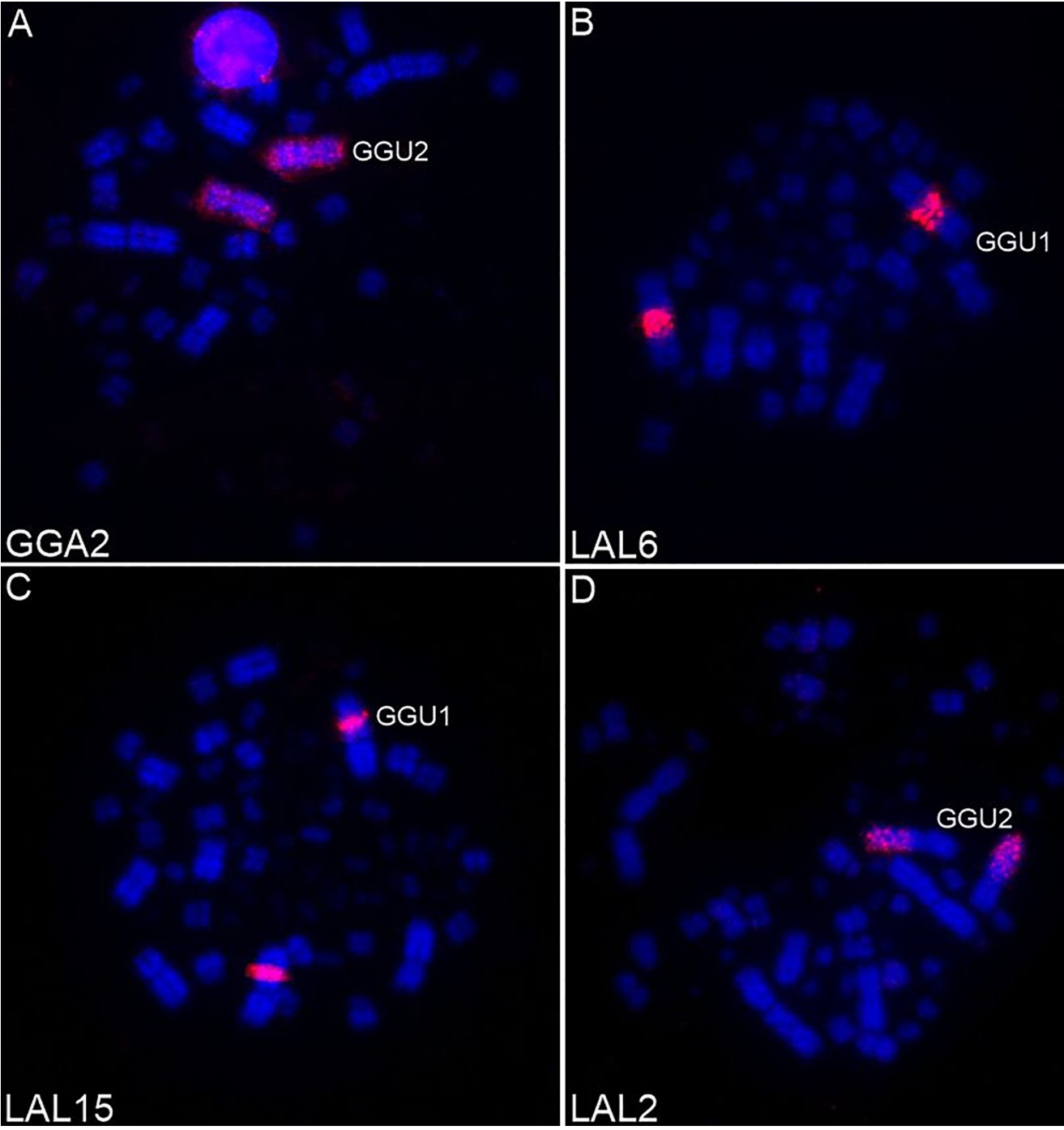

**Fig 3. Examples of fluorescence *in situ* hybridization experiments with whole-chromosome probe derived from *G. gallus* (GGA) and *L. albicollis* (LAL) onto *G. guira* (GGU) chromosomes.**

Moreover, the segment corresponding to GGA1qterm (PCA1q) was not hybridized by any LAL probes used, as well as in GGU. Additionally, LAL probes indicated that a pericentric inversion had occurred in PCA2 (GGA2q). A homology map based on the results of the experiments using GGA and LAL probes in PCA is shown in Fig 5B.

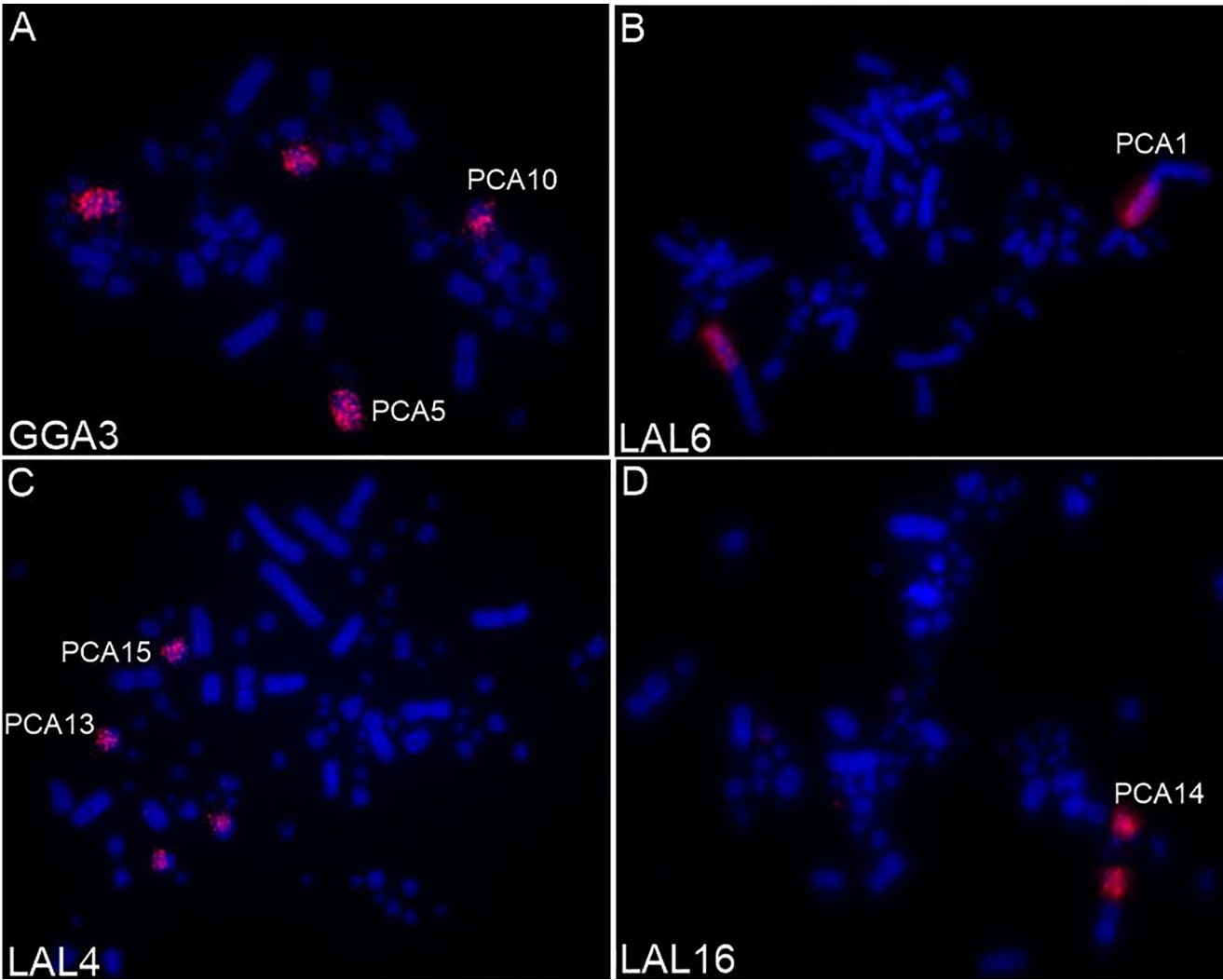

**Fig 4. Examples of fluorescence *in situ* hybridization experiments with whole-chromosome probe derived from *G. gallus* (GGA) and *L. albicollis* (LAL) onto *P. cayana* (PCA) chromosomes.**

## Discussion

This study contributes to an understanding of chromosome evolution in two cuckoo species by providing the first chromosome painting data for these species. Our results also demonstrate a high chromosomal diversity, with *G. guira* presenting 2n = 76 with fission and fusion events, while *P. cayana*, in contrast, presents mainly fissions and just one fusion event, which are responsible for the high diploid number found, 2n = 90. Interestingly, no chromosomal rearrangements were found in common between *G. guira* and *P. cayana*. Additionally, our results with Giemsa staining are compared to other cuckoos and taxa that have been proposed recently as sister groups to cuckoos (Otidiformes, Musophagiformes and Opisthocomiformes).

Cytogenetic data remain scarce for cuckoo species and are restricted to classical cytogenetics with conventional staining [14–19]. The diploid number for *G. guira* was first determined to be 2n = 72 by de Lucca [26] and 2n = 66 in the study of Waldrigues and Ferrari [27], while for *P. cayana*, it was reported to be 2n = 76 [14]. However, our results indicate that the diploid

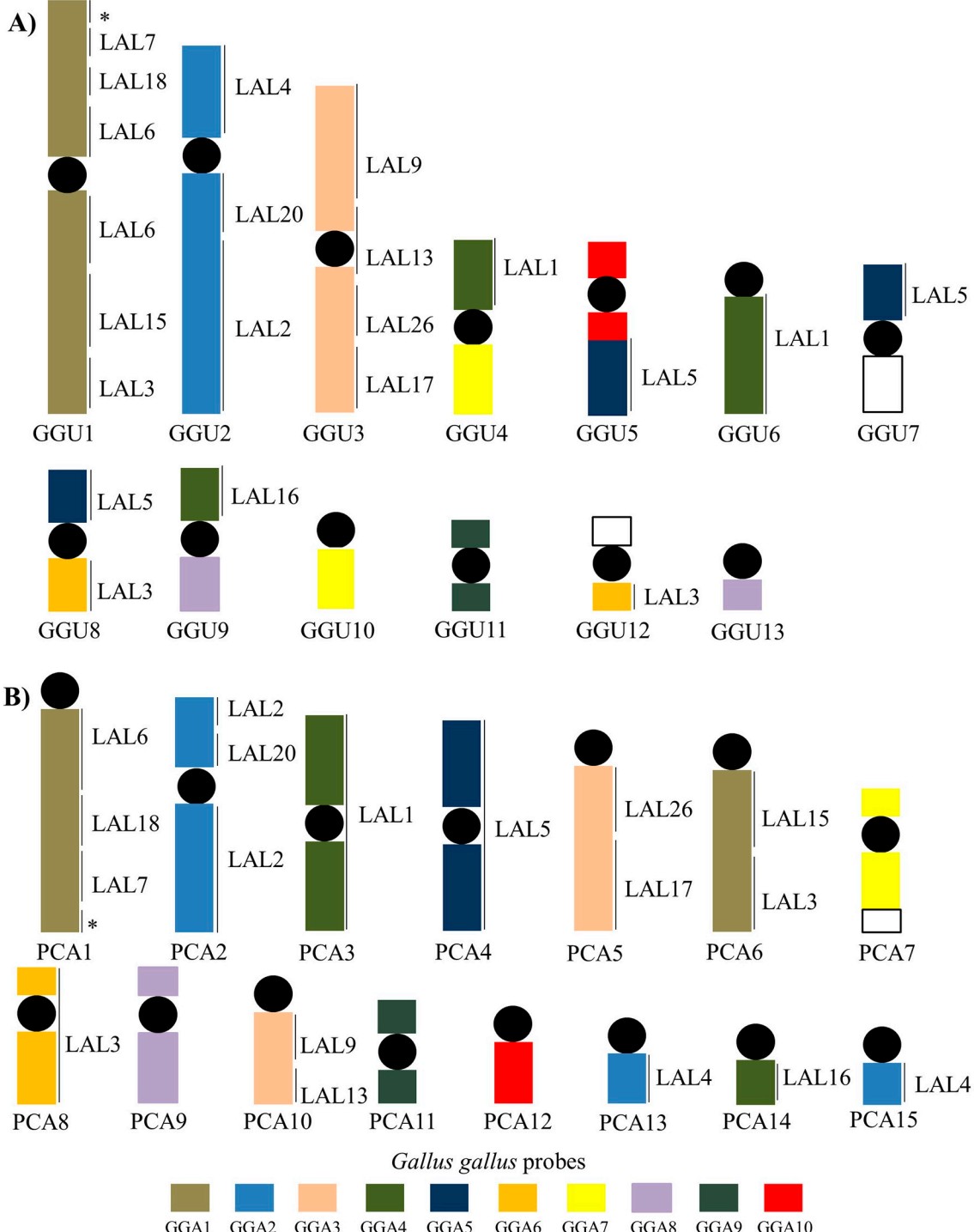

**Fig 5. Homologous chromosomal segments in *G. gallus* (GGA), *L. albicollis* (LAL) and *G. guira* (A, GGU), and *P. cayana* (B, PCA) macrochromosomes as detected by fluorescence *in situ* hybridization (FISH) using *G. gallus* and *L. albicollis* whole chromosome paints.** Segments not hybridized with LAL probes are indicated with an asterisk. Segments not hybridized with LAL and GGA probes are indicated by white gaps.

number for both species is higher, 2n = 76 for *G. guira* and 2n = 90 for *P. cayana*. These findings indicate that chromosomal information from historical publications needs to be reviewed.

Regarding chromosome morphology and size, three basic karyotype patterns could be observed in cuckoos: 1) The avian standard karyotype, as found in chicken (few macrochromosomes, the first eight pairs are clearly larger than the others), found in *Cuculus* species and in *Surniculus lugubris* [16,18,19]: 2) A karyotype in which the first three pairs are of similar size to those in group 1, but with a higher number of medium-size biarmed macrochromosomes, probably derived by Robertsonian translocation. These karyotypes are found in *G. guira* (present study) and in the genus *Crotophaga* [14,15] and: 3) A karyotype in which the first three pairs are clearly smaller than in other cuckoo species and in chicken, probably derived from chromosomal fissions. This pattern of chromosome organization can be found in *P. cayana* (present study), *Carpococcyx renauldi* (2n = 60) [17], and in *Phaenicophaeus tristis* (2n = 78) [19].

The chromosome painting results confirmed that ancestral syntenies corresponding to chicken chromosomes 1–3 are fully conserved in *G. guira* (Fig 5A), while in *P. cayana* karyotype (Fig 5B), these synteny groups have undergone fissions (Fig 7A). Besides that, chromosome fusions were observed in *G. guira*. Although we observed many fusions in *G. guira*, only one fusion was detected in *P. cayana*, involving the chromosome homologous to GGA7 and possibly a microchromosome (considering that this segment was not labeled by any of the chromosome paints used). As *G. guira* is in the same subfamily as *Crotophaga* species (Crotophaginae) and has a similar karyotype [14,15], we can predict that the chromosome rearrangements observed in *G. guira* might be similar in *Crotophaga* species.

The *L. albicollis* probes confirm the results obtained with *G. gallus* probes. However, in both species studied here the chromosomal reorganization would not have been found by the *G. gallus* probes alone. LAL6, which is homologous to a segment in GGA1q, revealed the occurrence of a pericentric inversion on chromosome 1 of *G. guira*, not found in *P. cayana* (Fig 6). This finding indicates that the inversion occurred after the divergence of the two species. Likewise, the fission of ancestral chromosome 1 in *P. cayana* also occurred after their divergence.

The centromeric fission of the chromosome homologous to GGA1 is a frequent event in birds, appearing independently in different orders [20]. Pair 2 of *P. cayana*, which is homologous to GGA2, also showed an inversion, which was observed with probes corresponding to LAL2 (Fig 7B). These findings are in accordance with the fact that intrachromosomal rearrangements are frequent in avian genomes and may contribute to the phenotypic diversity of different bird species [28,29].

When the cuckoo karyotypes are compared with those of turacos [30,31], a sister group to cuckoos in the analysis of Jarvis et al. [13], a drastic difference is clearly observed with karyotypes of Group 1 and 2 here proposed in Cuculiformes. However, turacos show karyotypes similar with group 3, because they are characterized by the first three pairs of similar size [30,31], and smaller than in cuckoos of Groups 1 and 2. The other difference between cuckoos from Group 1 and 2 and turacos relate to the diploid number. To our knowledge, only two turaco species have been karyotyped: *Musophaga violacea* and *Tauraco porphyreolophus*, both with 2n = 82, which are higher than in species from Groups 1 and 2, but not very different of Group 3.

The karyotype of the Houbara bustard (*Chlamydotis undulata*), the only species available in the order Otidiformes, is similar to the ancestral karyotype of birds with 2n = 78, with 8 pairs considered as macrochromosomes and 30 pairs as microchromosomes. Chromosome fissions were proposed in the chromosomes homologous to GGA4q and 5 based on G-banding

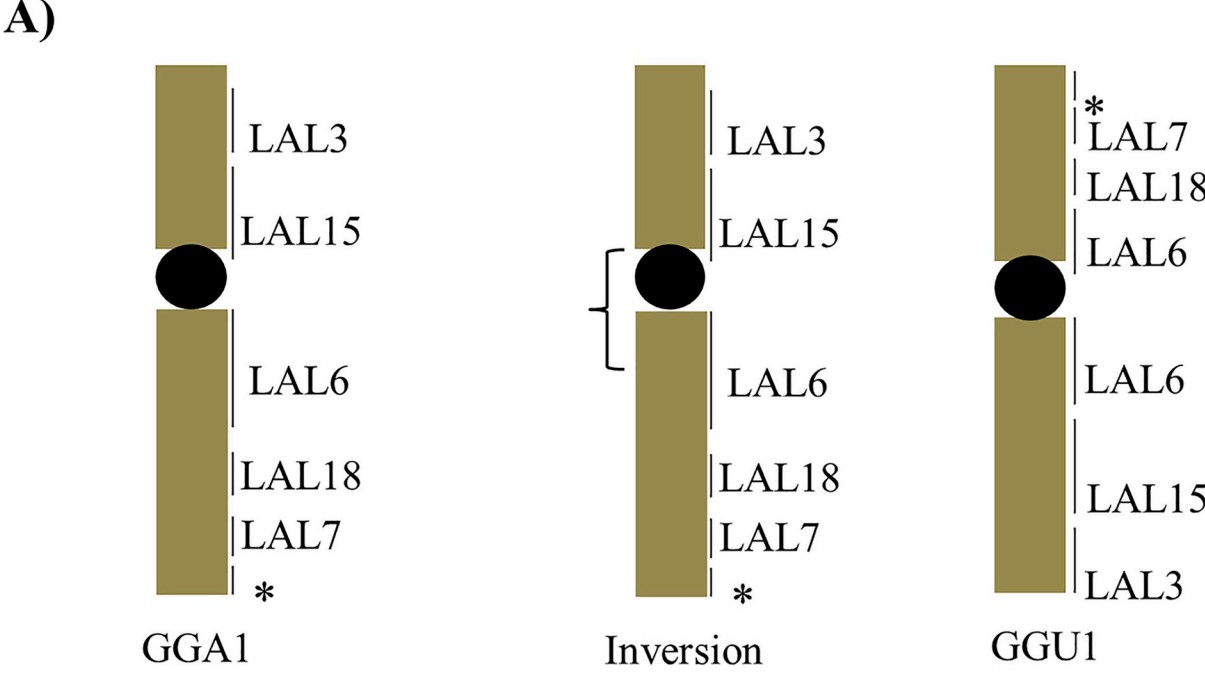

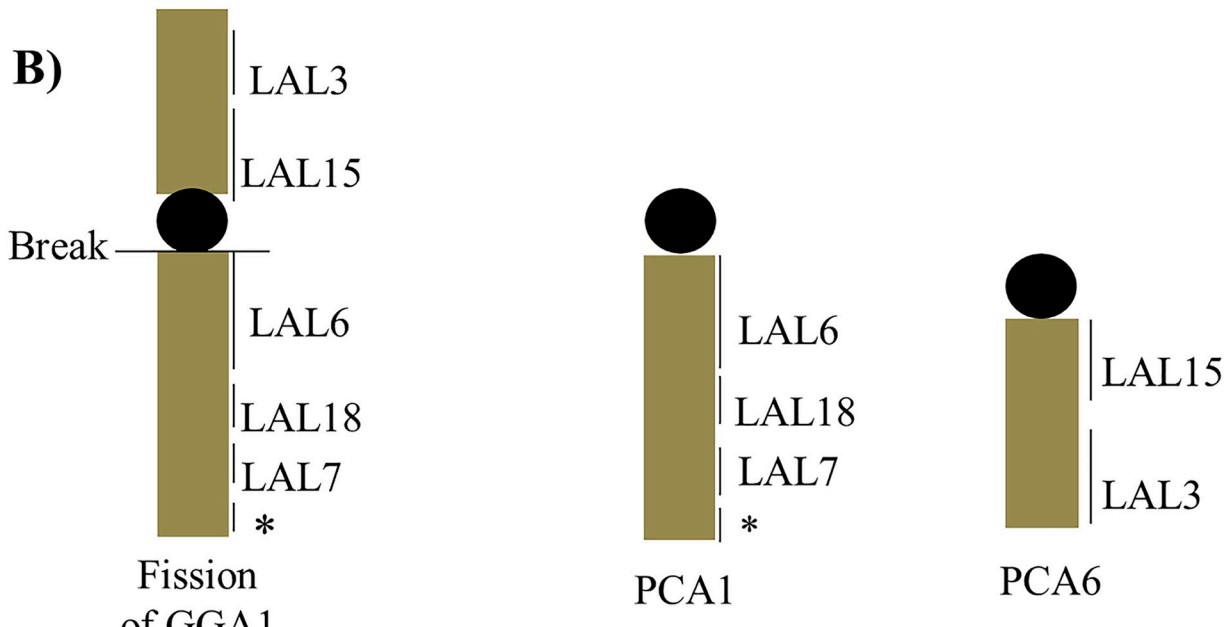

**Fig 6. Schematic representation of the structural rearrangements found in GGU1, PCA1 and PCA6.** GGA1q underwent a pericentric inversion, leading to GGU1 (A). A centric fission in the ancestral synteny homologous to GGA1 led to two chromosome pairs, homologous to GGA1p and GGA1q (B).

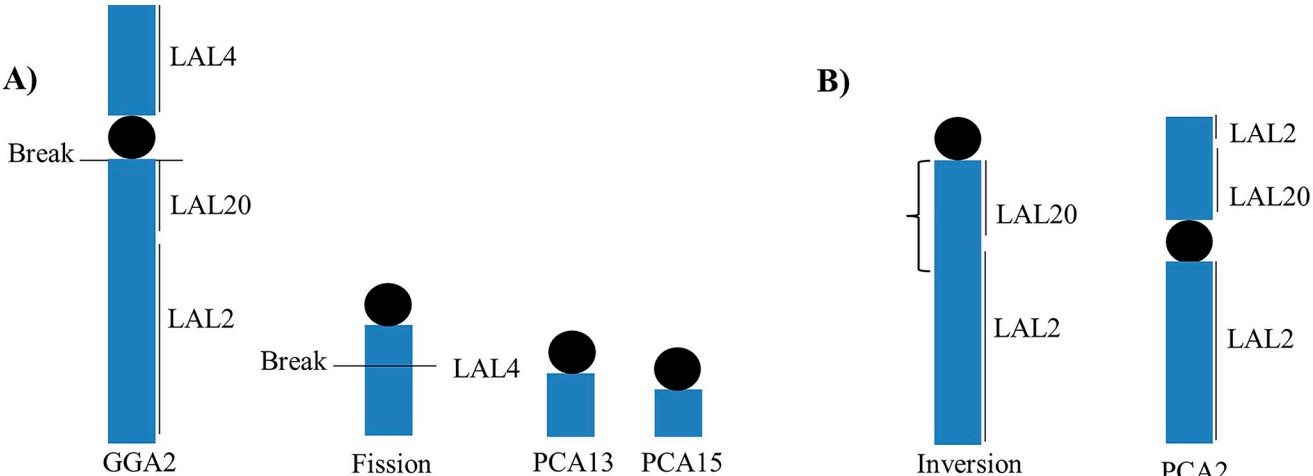

**Fig 7. Schematic representation of the structural rearrangements found in PCA2, PCA13, PCA15 and GGU3.** Two fissions in the ancestral synteny homologous to GGA2 led to two chromosome pairs, homologous to GGA2p (A) and one chromosome pair homologous to GGA2q. GGA2q underwent a pericentric inversion, forming PCA2 (B).

patterns [32]. If these rearrangements were confirmed by chromosome painting, they would represent synapomorphies with Cuculiformes, corresponding to the rearrangements observed in *G. guira*.

The hoatzin (*O. hoazin*) has been also considered as a sister taxon to cuckoos [33,34], or even considered to be a cuckoo [9]. However, the analysis performed by Hughes [8] support a sister relationship between the hoatzin and turacos, and not cuckoos. Sorenson and Payne [11] did not find evidence to include the hoatzin within Cuculiformes, or a sister relationship between cuckoos and hoatzin based on molecular data. Also, these authors did not find evidence to support a close relationship between turacos and cuckoos or between turacos and hoatzin. In a previous study, the fusion between GGA8/6 and GGA9/2 and the fissions of chromosomes homologous to GGA1 and GGA2 were found in the hoatzin (dos Santos et al., 2018). Although we found a fission in GGA1 in *P. cayana*, the breakpoints were not the same: in hoatzin the breakpoint was in an interstitial region, while in *P. cayana* it was a centric fission. However, the fission breakpoint on GGA2 in hoatzin was the same as in *P. cayana*. One of the resulting segments (corresponding to LAL4) was involved in a fusion event in hoatzin (GGA2q/10), while the same segment was involved in other fission in *P. cayana*.

In addition, the phylogenetic relationships within Cuculiformes are still problematic. Hence, *Piaya* occupies the basal position on the strict-consensus trees based on combined osteological, behavioral and ecological data, and based on behavioral and ecological data only [35]. However, a study based only on osteological characters shows a different topology in which the genus *Carpococcyx* occupies a basal position, while *Piaya* is in a more derived position, despite their similar karyotypes [8]. In despite of this, *Guira* and *Crotophaga*, which have similar karyotypes (Group 2) are placed as sister groups [8]. Therefore, although the importance of cytotaxonomy in supporting some phylogenetic results is clear, cytogenetic data of Cuculiformes are still fragmental, and only the analyses of a higher number of species can clarify the correct direction of rearrangements during the diversification of this order.

In conclusion, by combining comparative chromosome painting and conventional staining studies, we demonstrate that the cuckoo species can be classified into at least three groups based on their karyotypes. Unfortunately, so far, chromosome data have not defined the phylogenetic position and relationships of Cuculiformes with other birds. Considering that our

results show that this order exhibits clear karyological diversity, additional detailed studies on a higher number of species should help to clarify the phylogeny of this interesting group of birds.

## Acknowledgments

The authors would like to thank all the staff of the research group from the "Diversidade Genética Animal" from the Universidade Federal do Pampa and to the "Laboratório de Cultura de Tecidos e Citogenética", SAMAM, from the Instituto Evandro Chagas, for technical support. We would also like to thank the SISBIO for logistic support. This study was financed by Conselho Nacional de Desenvolvimento Científico e Tecnológico (CNPq), the Coordenação de Aperfeiçoamento de Pessoal de Nível Superior (CAPES) and Pró-Reitoria de Ensino e Pesquisa (PROPESP), Universidade Federal do Pará. Finally, we would like to thank Mr. Alex Araújo for the illustration of the G. guira used in Fig 1 and the P. cayana in Fig 2.

## Author Contributions

**Conceptualization:** Michelly da Silva dos Santos, Rafael Kretschmer, Edivaldo Herculano Corrêa de Oliveira.

**Data curation:** Michelly da Silva dos Santos, Rafael Kretschmer.

**Formal analysis:** Michelly da Silva dos Santos, Rafael Kretschmer, Ivanete de Oliveira Furo.

**Funding acquisition:** Malcolm A. Ferguson-Smith, Edivaldo Herculano Corrêa de Oliveira.

**Investigation:** Michelly da Silva dos Santos, Ivanete de Oliveira Furo, Edivaldo Herculano Corrêa de Oliveira.

**Methodology:** Michelly da Silva dos Santos, Ivanete de Oliveira Furo, Ricardo José Gunski, Mirela Pelizaro Valeri, Patricia C. M. O'Brien.

**Project administration:** Edivaldo Herculano Corrêa de Oliveira.

**Supervision:** Malcolm A. Ferguson-Smith, Edivaldo Herculano Corrêa de Oliveira.

**Writing – original draft:** Michelly da Silva dos Santos, Rafael Kretschmer.

**Writing – review & editing:** Michelly da Silva dos Santos, Rafael Kretschmer, Ricardo José Gunski, Analía del Valle Garnero, Patricia C. M. O'Brien, Malcolm A. Ferguson-Smith.

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
