## [Decision Letter · Decision Letter 0]

24 Mar 2020

PONE-D-20-07068

Chromosomal evolution and phylogenetic considerations in cuckoos (Aves, Cuculiformes, Cuculidae)

PLOS ONE

Dear Dr. de Oliveira,

Thank you for submitting your manuscript to PLOS ONE. After careful consideration, we feel that it has merit but does not fully meet PLOS ONE’s publication criteria as it currently stands. Therefore, we invite you to submit a revised version of the manuscript that addresses the points raised during the review process.

We would appreciate receiving your revised manuscript by May 08 2020 11:59PM. To enhance the reproducibility of your results, we recommend that if applicable you deposit your laboratory protocols in protocols.io, where a protocol can be assigned its own identifier (DOI) such that it can be cited independently in the future. For instructions see: http://journals.plos.org/plosone/s/submission-guidelines#loc-laboratory-protocols

We look forward to receiving your revised manuscript.

Kind regards,

Igor V. Sharakhov

Academic Editor

PLOS ONE

Journal Requirements:

2. In your Methods, please provide full details of the handling of the birds and methods used to obtain the biological samples used in your analyses.

Reviewers' comments:

Reviewer's Responses to Questions

**Comments to the Author**

1. Is the manuscript technically sound, and do the data support the conclusions?

Reviewer #1: Yes

Reviewer #2: Yes

2. Has the statistical analysis been performed appropriately and rigorously? 

Reviewer #1: N/A

Reviewer #2: N/A

3. Have the authors made all data underlying the findings in their manuscript fully available?

Reviewer #1: Yes

Reviewer #2: Yes

4. Is the manuscript presented in an intelligible fashion and written in standard English?

Reviewer #1: Yes

Reviewer #2: Yes

5. Review Comments to the Author

Reviewer #1: Dear Dr. de Oliveira,

I have carefully read your manuscript entitled "Chromosomal evolution and phylogenetic considerations in cuckoos (Aves, Cuculiformes, Cuculidae)". The manuscript contains new important information on karyotype structure and chromosomal evolution of two cuckoo species, Guira guira and Piaya cayana, using routine staining and whole chromosome painting with probes derived from two other bird species, and therefore this paper could be published in PLOS ONE. However, I believe that a few points should be addressed in the manuscript. First, I think that a brief characterization of the standard avian/chicken karyotype would be useful for those who are not closely familiar with cytogenetics of birds (see line 189). Second, you state that turacos have first three chromosome pairs of similar size, but these chromosomes are "larger" in the cuckoos of Groups 1 and 2 (see lines 232-233); if you in fact mean that the corresponding chromosomes of cuckoos are unequal in size, please rephrase the sentence. Third, if we assume that Piaya occupies the basal position on the phylogenetic tree due to certain morphological evidence, I do not think that morphological/karyotypic similarity of two other genera mentioned by you means close relationships between these groups, just because their characters may well represent respective symplesiomorphies (see lines 261-269). Moreover, I suggest that authors' names (and probably also years of first publication) are to be added to each species name mentioned both in the abstract and the main text for the first time. I also believe that the four main species names used in the text (i.e., Guira guira, Piaya cayana, Gallus gallus and Leucopternis albicollis) should be spelled out only if mentioned for the first time, and respectively cited as G. guira, P. cayana, G. gallus and L. albicollis in other parts of the text. Unnecessary text highlighting (lines 14 to 16) must be removed as well. Please change "round", "5", "cuckoos species" and "female" to "rounds", "5B", "cuckoo species" and "male" respectively (lines 97, 164, 195 and 406). Furthermore, please keep all your references in the reference list to the style recommended by PLOS, including removal of an unnecessary ISSN number (line 349). In addition, a few species/subspecies names given in the reference list (lines 318, 339 and 390) should be italicized.

Yours sincerely,

Reviewer #2: The presented manuscript aims to reconstruct the chromosome diversification in Cuculiformes using FISH with whole-chromosome probes in order to resolve the evolutionary processes in this taxon. This study is of importance as phylogenetic relationships of cuckoos is controversial and cytogenetic data are scarce, most of which require revision. The work is done at a good technical level and the manuscript meets the criteria for the publication in PLOS ONE journal.

Minor comments:

1) Lines 201-206. Consider revising the sentence: "In contrast to the extensive fusions observed in Guira guira, in Piaya cayana we observed just the fusion between the homologous chromosome GGA7 with an unidentified segment in PCA7, probably with a microchromosome, since none of the GGA and LAL probes tested – which corresponded to macrochromosome of GGA - hybridized to this segment"

2) Line 406. In the Figure 1 legend "a female Guira guira" should be replaced with "a male Guira guira"

3) Lines 410, 413. In the Figure 3 and 4 legends wcp (whole-chromosome probe) should be decrypted

4) Year is misssed in the reference 13

6. PLOS authors have the option to publish the peer review history of their article (what does this mean?). If published, this will include your full peer review and any attached files.

Reviewer #1: Yes: Vladimir E. Gokhman

Reviewer #2: No

---

## [Author Response · Author response to Decision Letter 0]

15 Apr 2020

Dear Reviewers,

First of all, we would like to thank you for taking your time and patience in order to try to make the manuscript better. We tried to follow most of your suggestions, and answer your doubts. We hope we have achieved these goals.

Sincerely,

The Authors

Reviewer 1

1. First, I think that a brief characterization of the standard avian/chicken karyotype would be useful for those who are not closely familiar with cytogenetics of birds (see line 189). 

A: Yes, we agree and we added a short description of the standar karyotype..

2. Second, you state that turacos have first three chromosome pairs of similar size, but these chromosomes are "larger" in the cuckoos of Groups 1 and 2 (see lines 232-233); if you in fact mean that the corresponding chromosomes of cuckoos are unequal in size, please rephrase the sentence.

A: Yes, we agree, the paragraph was unclear. We reformulated it.

3. Third, if we assume that Piaya occupies the basal position on the phylogenetic tree due to certain morphological evidence, I do not think that morphological/karyotypic similarity of two other genera mentioned by you means close relationships between these groups, just because their characters may well represent respective symplesiomorphies (see lines 261-269).

A: Yes, after a careful analysis, we decided to rewrite this paragraph, taking into account the phylogenetic proposal as a whole.

4. Moreover, I suggest that authors' names (and probably also years of first publication) are to be added to each species name mentioned both in the abstract and the main text for the first time. I also believe that the four main species names used in the text (i.e., Guira guira, Piaya cayana, Gallus gallus and Leucopternis albicollis) should be spelled out only if mentioned for the first time, and respectively cited as G. guira, P. cayana, G. gallus and L. albicollis in other parts of the text. Unnecessary text highlighting (lines 14 to 16) must be removed as well. Please change "round", "5", "cuckoos species" and "female" to "rounds", "5B", "cuckoo species" and "male" respectively (lines 97, 164, 195 and 406). Furthermore, please keep all your references in the reference list to the style recommended by PLOS, including removal of an unnecessary ISSN number (line 349). In addition, a few species/subspecies names given in the reference list (lines 318, 339 and 390) should be italicized.

A: Thank you for the corrections. We followed every corrections and suggestions, including the correct form of reference list recommended by Plos One. 

Reviewer 2

1. Lines 201-206. Consider revising the sentence: "In contrast to the extensive fusions observed in Guira guira, in Piaya cayana we observed just the fusion between the homologous chromosome GGA7 with an unidentified segment in PCA7, probably with a microchromosome, since none of the GGA and LAL probes tested – which corresponded to macrochromosome of GGA - hybridized to this segment"

A: Yes, the sentence was quite confuse. We changed it to “Although we observed many fusions in Guira gira, only one fusion was detected in Piaya cayana, involving the chromosome homologous to GGA7 and possibly a microchromosome (considering that this segment was not labeled by any of the chromosome paints used).

2. Line 406. In the Figure 1 legend "a female Guira guira" should be replaced with "a male Guira guira"

A: Yes, we corrected it.

3. Lines 410, 413. In the Figure 3 and 4 legends wcp (whole-chromosome probe) should be decrypted

A: Yes, we corrected it.

4. Year is misssed in the reference 13

A: Thank you very much, we corrected it.

---

## [Editor Report · Decision Letter 1]

17 Apr 2020

Chromosomal evolution and phylogenetic considerations in cuckoos (Aves, Cuculiformes, Cuculidae)

PONE-D-20-07068R1

Dear Dr. Oliveira,

We are pleased to inform you that your manuscript has been judged scientifically suitable for publication and will be formally accepted for publication once it complies with all outstanding technical requirements.

With kind regards,

Igor V. Sharakhov

Academic Editor

PLOS ONE
---

## [Editor Report · Acceptance letter]

24 Apr 2020

PONE-D-20-07068R1 

Chromosomal evolution and phylogenetic considerations in cuckoos (Aves, Cuculiformes, Cuculidae) 

Dear Dr. de Oliveira:

I am pleased to inform you that your manuscript has been deemed suitable for publication in PLOS ONE. Congratulations! Your manuscript is now with our production department. 

With kind regards,

on behalf of

Dr Igor V. Sharakhov 

Academic Editor

PLOS ONE